# Fatal Pneumonia Caused by *Beauveria bassiana* in a Kemp’s Ridley Sea Turtle (*Lepidochelys kempii*, Garman, 1880) on the Portuguese Coast: Case Report and Review of *Beauveria* spp. Infections in Reptiles

**DOI:** 10.3390/microorganisms13092092

**Published:** 2025-09-08

**Authors:** Gonçalo N. Marques, Ricardo Lopes, Maria Conceição Peleteiro, Jaqueline T. Bento, João R. Mesquita, Fábio Abade dos Santos, Leonor Delgado, Ana Cláudia Coelho, Miguel Lourenço, Miriam Leal, Virgínia Lopes, Ana Paula Castro, Rita Barny, Joana Guerra, Nuno Urbani, Antonieta Nunes, Yohann Santos, Isabel Gaspar, Andreia Garcês, João Neves

**Affiliations:** 1Zoomarine Portugal, E.N. 125 Km 65, Guia, 8201-864 Albufeira, Portugal; goncalo.marques@zoomarine.pt (G.N.M.); miguel.lourenco@zoomarine.pt (M.Lo.); ritab@zoomarine.pt (R.B.); joana.silva@zoomarine.pt (J.G.); nuno.urbani@zoomarine.pt (N.U.); antonieta.nunes@zoomarine.pt (A.N.); yohann.santos@zoomarine.pt (Y.S.); isabel.gaspar@zoomarine.pt (I.G.); joao.neves@zoomarine.pt (J.N.); 2Department of Veterinary Sciences, University of Trás-os-Montes and Alto Douro, 5000-801 Vila Real, Portugal; lopes.rmv@gmail.com; 3CEDIVET Veterinary Laboratories, Lionesa Business Hub, R. Lionesa 446 C24, 4465-671 Leça do Balio, Portugal; 4Animal and Veterinary Sciences Department, University Institute of Health Sciences, Advanced Polytechnic and University Cooperative-CESPU, CRL, 1317, 4585-116 Gandra, Portugal; leonordelgado@inno.pt; 5URANOLABPT, Avenida Pedro Álvares Cabral, Centro Empresarial Sintra Estoril VI V E23, 2710-297 Sintra, Portugal; 6School of Medicine and Biomedical Sciences (ICBAS), University of Porto, 4050-313 Porto, Portugal; jtbento@icbas.up.pt (J.T.B.); gilopes61@gmail.com (V.L.);; 7Centro de Estudos de Ciência Animal (CECA), Instituto de Ciências, Tecnologias e Agroambiente (ICETA), University of Porto, 4051-401 Porto, Portugal; 8Associate Laboratory for Animal and Veterinary Science (AL4AnimalS), 1300-477 Lisboa, Portugal; accoelho@utad.pt; 9Laboratório de Virologia Animal, INIAV, Avenida da República, Quinta do Marquês, 2780-157 Oeiras, Portugal; fabio.abade@iniav.pt; 10INNO Veterinary Laboratories, Rua Cândido de Sousa 15, 4710-300 Braga, Portugal; 11GIPOC (UNIPRO)-Comparative Oral Pathology Research Group, University Institute of Health Sciences, Advanced Polytechnic and University Cooperative-CESPU, CRL, 1317, 4585-116 Gandra, Portugal; 12Veterinary and Animal Research Centre (CECAV), Universidade de Trás-os-Montes e Alto Douro, Quinta de Prados, 5000-801 Vila Real, Portugal; 13Research in Veterinary Medicine (I-MVET), Faculty of Veterinary Medicine, Lusófona University, Lisbon University Centre, Campo Grande, 376, 1749-024 Lisbon, Portugal; 14Veterinary and Animal Research Centre (CECAV), Faculty of Veterinary Medicine, Lusófona University, Lisbon University Centre, Campo Grande, 376, 1749-024 Lisbon, Portugal; 15Unidade Local de Saúde de Santo António (ULS de Santo António), Largo Prof. Abel Salazar, 4099-001 Porto, Portugal; 16Wildlife Rehabilitation Center (CRAS-UTAD), Teaching Hospital, University of Trás-os-Montes and Alto Douro, Quinta de Prados, 5000-801 Vila Real, Portugal

**Keywords:** *Beauveria bassiana*, fungi, Kemp’s ridley sea turtle, *Lepidochelys kempii*, pneumonia, Portugal

## Abstract

The Kemp’s ridley sea turtle (*Lepidochelys kempii*) is the most critically endangered sea turtle species, with a distribution primarily restricted to the Gulf of Mexico. Its occurrence along the Iberian Peninsula is exceedingly rare. This study describes the postmortem findings of a juvenile *L. kempii* rescued off the Portuguese coast in 2024, which died after 11 days in rehabilitation despite intensive supportive care. Necropsy revealed severe, diffuse pneumonia. Histopathological examination showed interstitial inflammation and branching septate hyphae, while fungal culture and DNA sequencing confirmed the presence of *Beauveria bassiana*. Mycotic diseases in reptiles are often underrecognised but can lead to significant morbidity and mortality, particularly in immunocompromised or stressed individuals such as stranded marine turtles. This rare occurrence of a *L. kempii* on the Portuguese coast provides important insights into the species’ dispersal patterns and underlines the potential conservation implications of opportunistic fungal infections in endangered species.

## 1. Introduction

The Kemp’s ridley sea turtle (*Lepidochelys kempii*, Garman, 1880) is the most endangered sea turtle species worldwide and is classified as “Critically Endangered” by the IUCN Red List of Threatened Species [1,2]. *L. kempii* has a restricted geographic distribution, concentrated mainly in the Gulf of Mexico, with the primary nesting sites located along the Mexican coast of Tamaulipas [3,4]. Although both adult and juvenile individuals inhabit the Gulf year-round, some juveniles are occasionally transported into the northwestern Atlantic Ocean by prevailing currents, with sporadic records as far north as Nova Scotia and, more rarely, in European waters [1,5,6]. In the Iberian Peninsula, confirmed sightings of *L. kempii* are exceptionally rare. Up until 2014, only a few individuals had been recorded off the coasts of Spain and Portugal [7,8,9,10]. A small cluster of juvenile turtles observed in 2014 raised interest in possible shifts in oceanographic patterns or dispersal routes [1].

Infectious diseases are key contributors to morbidity and mortality in sea turtles, particularly in individuals undergoing environmental stress, injury, or rehabilitation [11,12]. While bacterial infections have been extensively characterised, fungal diseases in sea turtles remain underdiagnosed and poorly understood [12]. Several fungal pathogens have been described in pulmonary infections in Kemp’s ridley sea turtles, including *Purpureocillium, Fusarium*, *Trichoderma*, *Beauveria*, *Candida*, *Penicillium*, *Mucor*, and *Colletotrichum* spp. [12,13,14,15].

Entomopathogenic fungi from the genera *Beauveria* and *Metarhizium* have emerged as significant opportunistic pathogens causing fatal pulmonary mycoses in reptiles, particularly marine turtles [15]. These fungi, traditionally recognised for their insecticidal properties in biological control systems, now represent a paradigm shift in reptilian pathology, demonstrating their capacity to establish severe respiratory infections across multiple species including loggerhead sea turtles (*Caretta caretta*), green sea turtles (*Chelonia mydas*), Kemp’s ridley sea turtles (*L. kempii*), and various terrestrial reptilian species [15,16]. The clinical presentation is usually characterised by angioinvasive granulomatous pneumonia, with distinctive pathological features including extensive tissue necrosis, vascular invasion, and disseminated organ involvement [15,17]. Histopathological examination reveals severe heterophilic to granulomatous inflammation with intralesional fungal elements, frequently accompanied by necrotising pneumonia and extensive pulmonary consolidation [12].

This case describes a rare occurrence of a juvenile *L. kempii* off the Portuguese coast, highlighting its ecological significance and contributing to knowledge of the species’ dispersal patterns. This report also represents the first documented case in Portugal of pneumonia in *L. kempii* caused by *Beauveria bassiana*. Further assessment of the in vitro antifungal susceptibility of *B. bassiana* provides preliminary data on potential therapeutic options for managing such infections. Additionally, the authors present a summary of *B. bassiana* infections in reptiles. By integrating clinical, microbiological, and ecological perspectives, this work provides practical insights in a rehabilitation context and highlights the growing relevance of fungal pathogens in sea turtle conservation. The alarming conservation status of Kemp’s ridley turtles further underscores the broader significance of these findings. It reinforces the urgent need to incorporate mycological diagnostics and antifungal susceptibility testing into rehabilitation protocols for endangered sea turtles. Additionally, this information is important because these pathogens have zoonotic potential.

## 2. Materials and Methods

### 2.1. Review on Beauveria bassiana Infections in Reptiles

A comprehensive literature review was conducted to identify documented cases of *Beauveria bassiana* infections in reptiles, including both wild and captive individuals. The search focused on identifying relevant peer-reviewed publications, case reports, and scientific communications that provided detailed information on host species, clinical presentation, diagnosis, and outcomes. Search terms included combinations of keywords such as “*Beauveria bassiana*”, “*Beauveria* spp.”, “fungi”, “mycosis”, “reptiles”, “turtles”, “snakes”, “lizards”, and “crocodiles”. Digital academic databases used for the search included ResearchGate, Google Scholar, and PubMed. Articles were screened for relevance, and only those that contained explicit data on fungal infection in reptiles, including species identification, pathological findings, and clinical outcome, were included in the final review.

### 2.2. Clinical Case: Kemp’s Ridley Sea Turtle

In May 2024, a juvenile Kemp’s ridley sea turtle was recovered off the coast of Sines, Portugal. The specimen was subsequently transported to Porto d’Abrigo, the wildlife rehabilitation centre of Zoomarine, Portugal, for clinical evaluation and treatment. Upon admission, standard intake protocols were followed, including biometric measurements, physical examination, blood analyses, and diagnostic imaging. Blood samples were obtained from the dorsal cervical vessels and immediately transferred into lithium heparin tubes. Both total erythrocyte and leukocyte counts were performed manually, in which a 5 µL blood-filled pipette was inserted into a Natt-Herricks-TIC^®^ (Bioanalytic GmbH, Umkirch, Germany) 1:200 stain solution vial, and counting was performed with a Neubauer chamber. Evaluation of blood smears was performed after Diff-Quik (MAIM S.L., Barcelona, Spain) staining. Haemoglobin levels were obtained through a haemoglobin analyser (HemoCue, Ängelholm, Sweden), and haematocrit values were obtained after centrifugation of microhaematocrit tubes (Centurion Scientific Ltd.-Pro-Vet, Sussex, UK; 12,000 rpm, 5 min). The biochemical profile was analysed using a VETSCAN^®^ VS chemistry analyser (Avian/Reptilian Profile Plus–ABAXIS Europe GmbH, Griesheim, Germany). Radiographic examination was performed with portable radiographic equipment (GIERTH HF300, GmbH, Neu-Isenburg, Germany). Supportive care and monitoring were provided according to established rehabilitation procedures. Despite intervention, the animal’s condition deteriorated, and it died on day 11 post-admission.

#### 2.2.1. Postmortem Examination and Sample Preparation

A complete necropsy was conducted shortly after the animal’s death. Representative tissue samples were collected and fixed in 10% neutral buffered formalin for histopathological evaluation [18]. The samples were routinely processed, embedded in paraffin wax, sectioned at 3 μm thickness, and stained with haematoxylin and eosin (H&E). Lung tissue sections were additionally stained with periodic acid–Schiff (PAS) to assess for fungal elements and mucopolysaccharides. Sterile swabs from the lungs and coelomic cavity were collected for microbiological analysis under aseptic conditions. For virological investigation, frozen tissue samples from the liver and lungs were submitted for PCR testing targeting herpesvirus, adenovirus, and paramyxovirus.

#### 2.2.2. Microbiological Analysis and Antifungal Susceptibility Testing

Lung tissue samples were directly inoculated onto Sabouraud Dextrose Chloramphenicol Agar (Frilabo, Maia, Portugal), incubated at both 25 °C and 37 °C, and observed daily for the development of fungal colonies. Phenotypic identification of fungal isolates was based on both macroscopic and microscopic characteristics. Macroscopic evaluation included assessment of colony growth rate, topography, texture, surface and reverse pigmentation, and the presence and colour of any diffusible pigment. Microscopic identification was performed by examining hyphal features—colour, size, septation, and structure—and conidial characteristics, including septation, shape, size, colour, wall texture, arrangement, and the morphology and development of conidiogenous cells, using Lactophenol Cotton Blue stain (Labkem, Barcelona, Spain) [19,20].

Antifungal susceptibility testing of the fungal isolate was performed using the E-test (MIC Test Strip, Liofilchem^®^, Roseto degli Abruzzi, Italy) according to the manufacturer’s instructions and following the guidelines of the Clinical and Laboratory Standards Institute (CLSI) M38-A2 for filamentous fungi. A conidial suspension was prepared from a fresh culture grown on Gelose RPMI (bioMérieux, Marcy-l’Étoile, France) and adjusted to a turbidity equivalent to 0.5 McFarland standard. The suspension was evenly spread on the surface of RPMI 1640 agar medium supplemented with 2% glucose and buffered to pH 7.0 with 0.165 M MOPS. E-test strips containing a predefined gradient of antifungal agents (amphotericin B, voriconazole, fluconazole, itraconazole) were applied to the agar surface, and plates were incubated at 35 °C for 24–48 h, depending on fungal growth rate. The minimum inhibitory concentration (MIC) was determined as the point of intersection between the elliptical zone of inhibition and the E-test strip scale. Results were interpreted based on CLSI [21] breakpoints.

#### 2.2.3. Molecular Analysis of Fungi

For molecular fungal identification, DNA extraction was performed by scraping off the mycelium from the cultured colonies and diluting it (10%) in ATL buffer. The mixtures were incubated at 100 °C for 10 min, followed by centrifugation at 8000× *g* for 5 min. After centrifugation, a 140 μL aliquot of the supernatant was used for DNA extraction and purification using the QIAamp DNA Mini Kit (Qiagen, Hilden, Germany), according to the manufacturer’s instructions. Automated extraction was performed using the QIAcube platform (Qiagen). The internal transcribed spacer (ITS) region and the 18S region of fungal rDNA were amplified using the primers ITS1 (5′-TCCGTAGGTGAACCTGCGG-3′) and ITS4 (5′-TCCTCCGCTTATTGATATGC-3′) [22] and SSU-F1 (5′-AACCTGGTTGATCCTGCCAGTAGTC-3′) and SSU-R1 (5′-TGATCCTTCTGCAGGTTCACCTACG-3′) [23].

PCR reactions were performed on a thermocycler (Bio-Rad, Hercules, CA, USA). Reaction mixtures were prepared using SpeedySupreme NZYTaq 2x Green Master Mix (NZYTech, Lisbon, Portugal) following the manufacturer’s instructions. PCR conditions included an initial denaturation at 95 °C for 5 min, followed by 35 cycles at 94 °C for 30 s, 55 °C for 30 s, and 72 °C for 1 min, ending with a final extension at 72 °C for 10 min. Following PCR amplification, DNA fragments were separated by electrophoresis on 1.5% agarose gels stained with Xpert Green Safe DNA gel dye (GRiSP^®^, Porto, Portugal). The electrophoresis was run at a constant voltage of 120 V for 25 min. The results were visualised by exposing the gels to UV light. PCR products were purified using the GRS PCR and Gel Band Purification Kit (GRiSP^®^). After purification, bidirectional sequencing was performed with the Sanger method using the appropriate internal primers for the target gene. Sequences were then aligned using the BioEdit Sequence Alignment Editor v7.1.9 software package, version 2.1v, and compared with those in the NCBI (GenBank) nucleotide database: https://blast.ncbi.nlm.nih.gov/Blast.cgi?PROGRAM=blastn&PAGE_TYPE=BlastSearch&LINK_LOC=blasthome (accessed on 1 June 2025). The phylogenetic relationships of the *Beauveria bassiana* isolates obtained in this study were assessed in comparison with reference sequences from GenBank, along with their respective accession numbers. The phylogenetic analysis was conducted in MEGA X using the maximum-likelihood approach (T92+G model). The resulting tree was refined and visualised using the Interactive Tree of Life (iTOL) platform.

#### 2.2.4. Virological Analysis

For virological analysis, lung and liver samples were homogenised separately at 20% (*w*/*v*) in phosphate-buffered saline (PBS) and clarified by centrifugation at 3000× *g* for 5 min. Total DNA and RNA were extracted from 200 μL of the clarified supernatants using the MagAttract 96 cador Pathogen Kit on a BioSprint 96 nucleic acid extractor (Qiagen, Hilden, Germany), following the manufacturer’s instructions. A 5 μL aliquot of total nucleic acids was used in PCR reactions, including a generalist nested PCR targeting the herpesviral DNA polymerase, which enables detection of herpesviruses from different subfamilies [24], a PAN-Paramyxoviridae protocol [25], and a general adenovirus detection method [26]. All PCR reactions were performed using a BIO-RAD PCR Thermal Cycler, Model T100. Herpesvirus and adenovirus amplification used NZYTaq II 2× Green Master Mix with the following cycling conditions: 96 °C for 2 min, 40 cycles of 96 °C for 30 s, 46 °C for 40 s, and 72 °C for 1 min, followed by a final extension at 72 °C for 7 min. Paramyxoviridae detection was performed using the QIAGEN OneStep RT-PCR Kit with the following protocol: 50 °C for 30 min (reverse transcription), 94 °C for 2 min, 40 cycles of 94 °C for 15 s, 48 °C for 30 s, and 72 °C for 30 s, with a final extension at 72 °C for 7 min. All PCR kits were used according to the manufacturer’s protocols.

## 3. *Beauveria bassiana* Infections in Reptiles

*Beauveria bassiana* is an entomopathogenic fungus naturally occurring in soils worldwide. It belongs to the Hypocreales order, Cordycipitaceae family, and *Beauveria* genus [27]. First identified by Agostino Bassi in 1835 in silkworms (*Bombyx mori*), molecular studies have revealed that it is a species complex comprising multiple phylogenetically distinct lineages [28]. It is also the anamorph (asexual form) of *Cordyceps bassiana*, the teleomorph known only from eastern Asia [29].

*B. bassiana* infects various arthropods, causing white muscardine disease [30], characterised by fungal overgrowth emerging from host cadavers. Infection occurs when spores penetrate an insect’s cuticle, proliferate internally, and lead to host death within days [31]. Fungal growth then emerges externally to release more spores [32] (Figure 1). This fungus is widely used as a biological insecticide against pests such as aphids, beetles, thrips, termites, and whiteflies, and is under study for control of bed bugs and malaria-transmitting mosquitoes [33].

Though *B. bassiana* is primarily recognised as an entomopathogenic fungus, it has increasingly been reported as an opportunistic pathogen in reptiles and other vertebrates. These infections are rare but often severe, with high mortality rates [15,29,31].

Table 1 summarises the documented cases of *B. bassiana* infections in various reptile species, including turtles, tortoises, crocodilians, lizards, and snakes. Lesions range from localised skin involvement to disseminated systemic infections, with the lungs being the most commonly affected organ. Table 1 highlights that most infections led to death, particularly when the respiratory tract was involved. Pulmonary lesions were frequently characterised by granulomatous or pyogranulomatous inflammation, pleuritis, pneumonia, or the replacement of normal lung tissue with consolidated, necrotic material. In some cases, coelomic involvement (e.g., hepatitis) or skin lesions were also noted. Diagnosis was typically confirmed through histopathology and fungal culture or molecular identification [15].

Although the exact pathogenesis in ectothermic vertebrates remains unclear, several predisposing factors have been suggested. Environmental exposure to contaminated substrates, compromised immunity, and the reptile’s thermoregulatory biology are considered major contributors [15]. In captive and rehabilitative settings, reptiles often face multiple stressors, such as inadequate temperatures, poor nutrition, and/or concurrent illnesses, that impair their immune responses, making them more susceptible to opportunistic fungal infections [12]. Notably, *B. bassiana* grows optimally at moderate temperatures (25–30 °C), a range which overlaps with the preferred thermal zones of many reptilian species [39,40]. Thus, when reptiles are housed below or at the lower end of their thermoneutral range, it may facilitate fungal proliferation. Cases in American alligators (*Alligator mississippiensis*) [34], Galápagos tortoises (*Testudo elephantopus*) [36], red-eared sliders (*Trachemys scripta*) [39], and multiple sea turtle species have demonstrated fungal colonisation and tissue invasion even in the absence of insect hosts, supporting a primary pathogenic role for *B. bassiana* under certain conditions [33].

Histologically, these infections often display extensive granulomatous inflammation, with abundant hyphae and conidia observed in tissue sections [34]. In advanced infections, widespread tissue necrosis and secondary bacterial infections may also occur, complicating diagnosis and treatment [16,37]. Since fungal infections in reptiles can develop insidiously, they are often diagnosed only in terminal stages or postmortem, underscoring the need for increased awareness, early detection, and appropriate antifungal strategies in herpetological medicine [34].

## 4. Clinical Case: Kemp’s Ridley Sea Turtle

A juvenile Kemp’s ridley sea turtle (*Lepidochelys kempii*) rescued off the Portuguese coast was admitted to Porto d’Abrigo (Zoomarine). Upon admission, the animal weighed 2.023 kg, with a straight carapace length (SCL) of 24.4 cm and a calculated body condition index (BCI) of 1.30, using Fulton’s condition factor ([weight (kg)/SCL^3^ (cm)] × 10^4^) as described by Bjorndal et al. (2000) [41]. The turtle appeared lethargic and displayed multiple lesions on the head and carapace (Figure 2), with a substantial load of epibionts.

The turtle was initially housed in a low-level freshwater tank for 24 h to facilitate monitoring, and was then transferred to a 2 m-diameter, 1.9 m-deep seawater tank (6 m^3^ capacity) maintained at 23–25 °C. Water quality parameters were monitored and adjusted according to standards for marine turtle rehabilitation [42], including filtration and chlorination systems.

Comprehensive medical care involved physical examinations, serial blood analyses (haematology and biochemistry), radiographic imaging, and microbiological cultures of the carapace lesions. Haematological data revealed a progressive decline in haematocrit levels (day 0: 28%; day 10: 7%) and hypoalbuminemia (day 0: 0.6 g/dL; day 10: 0.2 g/dL). Elevated levels of creatine kinase (CK) and aspartate aminotransferase (AST) were observed (CK—day 0: 5852 U/L; day 10: 2346 U/L and AST—day 0: 268 U/L; day 10: 417 U/L), along with persistent hyperphosphatemia (day 0: 11.2 mg/dL; day 10: 9.0 mg/dL). Notably, sustained hyperglycaemia was documented: 151 mg/dL (day 0); 144 mg/dL (day 2); 256 mg/dL (day 3); 225 mg/dL (day 4); 251 mg/dL (day 7); and 332 mg/dL (day 10).

Radiographic evaluation showed diffuse increased pulmonary opacity, most pronounced in the right lung, along with ill-defined focal opacities in the left lung field (Figure 3). Microbiological cultures of the antemortem carapace lesions yielded *Shewanella algae*, *Citrobacter gillenii*, and *Mucor* spp.

The therapeutic protocol included normobaric oxygen therapy, fluid support, and administration of marbofloxacin (2 mg/kg IM, SID), ketoprofen (2 mg/kg SC, SID), tramadol (5 mg/kg IM, every other day), iron dextran (5 mg/kg SC, twice weekly), and inhalation therapy with nebulised F10 SC^®^ solution (1:125 dilution, 30 min, BID). Despite intensive medical management, the turtle’s clinical condition deteriorated progressively, and the animal died on day 11 of hospitalisation.

Postmortem findings included diffuse areas of pulmonary congestion and oedema, along with multiple firm and whitish coalescing granules in the lungs, from 0.1 to 2.0 cm in diameter (Figure 4). These lesions had a dry, powdery consistency on the cut surface. No significant gross lesions were observed in other organs. Histopathological examination of the lungs revealed that approximately 60% of the tissue was affected by severe heterophilic bronchial and interstitial pneumonia (Figure 5). Bronchial and faveolar lumens were filled with mucinous material, sloughed pneumocytes (both intact and necrotic), numerous heterophils, bacterial aggregates (predominantly thin, short rods), and abundant septate fungal hyphae. PAS staining confirmed the fungal elements, which were observed in large numbers in some areas, associated with wide spaces which could represent the rupture of some faveoli. Lymphocytic infiltration was also noted between pneumocytes. Additional findings in other organs included diffuse hepatocellular vacuolisation and low numbers of melanomacrophages, particularly in the spleen and liver.

Fungal culture of lung samples incubated at 25 °C produced rapid growth of cottony, white colonies with slightly irregular margins by day 4 (Figure 6A). The reverse side of the colonies was non-pigmented. Micromorphology included narrow, septate hyaline hyphae and small, single-celled conidia arranged in dense clusters (Figure 6B), consistent with *Beauveria* spp. No fungal growth was observed at 37 °C.

Antifungal susceptibility testing was performed using the E-test method (MIC Test Strip, Liofilchem^®^, Roseto degli Abruzzi, Italy). The minimum inhibitory concentrations (MICs) obtained were as follows: fluconazole > 256 µg/mL (resistant), voriconazole 0.004 µg/mL (apparently susceptible), itraconazole 0.047 µg/mL (apparently susceptible), and amphotericin B > 256 µg/mL (resistant).

Bacterial cultures from lung tissue were also performed, showing the presence of *Vagococcus fluvialis*, *Pseudomonas putida*, and *Citrobacter gillenii. V. fluvialis* and *Aeromonas media* were isolated from coelomic cavity samples. All bacterial isolates were sensitive to marbofloxacin, with the exception of *V. fluvialis.*

Concerning molecular fungal identification, BLAST analysis of the 18S rRNA region (sequence PV39115) confirmed the isolate as belonging to the genus *Beauveria*. Further sequencing of the ITS rRNA region (sequence PV368950) demonstrated 100% identity with *B. bassiana* (reference sequence MZ956767).

Virological screening of liver and lung samples for herpesvirus, adenovirus, and paramyxovirus by PCR was performed in duplicate and yielded negative results.

## 5. Discussion

The presence of Kemp’s ridley sea turtles (*L. kempii)* in Iberian waters remains a rare event, reflecting the species’ historically restricted distribution, primarily within the Gulf of Mexico [43]. Caillouet and Gallaway (2020) analysed the movement pattern of these turtles, describing that juveniles in the Gulf can be transported by the Loop Current through the Florida Straits to the western North Atlantic [5]. Ocean circulation models indicate that between 5.1 and 28.4% of individuals in the ocean phase are swept out of the Gulf to the northwest Atlantic coast annually. From there, the Gulf Stream can propel these juveniles further north, to the Canadian coast, or eastwards through the North Atlantic Gyre, potentially reaching European waters, including the Portuguese coast [3,5]. However, this natural dispersal mechanism represents a loss to the Gulf of Mexico population, as most turtles transported to European waters may not return to their area of origin, potentially affecting its population growth and recovery [3,5,44]. The increasing number of Kemp’s ridley sea turtles in Europe may reflect changes in population dynamics or environmental conditions [1,44], involving such factors as the recovery of nesting populations following conservation efforts or environmental shifts altering oceanic drift patterns [1,45,46]. However, the survival of these turtles in non-native environments remains uncertain [6].

Avens and Dell’Amico (2018) [44] concluded that Kemp’s ridley sea turtles in the French Atlantic showed lower growth rates and a smaller size-at-age compared to those in their typical distribution area. This suggests that environmental conditions in Atlantic waters may be sub-optimal for the species. In the present case, the reported sea surface temperature (SST) at the date of rescue was approximately 15 °C, well below the thermal preference for this species. In addition, although the animal presented in this work was rescued in May, the winter sea temperature in Iberian waters can drop below 10 °C, possibly triggering cold-stunning phenomena in sea turtles and increased susceptibility to disease [1]. In this rescued individual, prolonged exposure to cold, combined with the energetic cost of the trans-Atlantic drift, likely contributed to chronic debilitation, malnutrition, and immune suppression, predisposing it to opportunistic infections.

One such infection was attributed to *Beauveria bassiana*, a filamentous entomopathogenic fungus widely distributed in terrestrial and aquatic environments [47]. Although typically pathogenic to insects, *B. bassiana* has been increasingly reported as an opportunistic pathogen in reptiles, including turtles, alligators, lizards, and snakes [13,34,39]. In reptiles, *B. bassiana* infections are associated with high mortality and frequently occur in conjunction with immunosuppressive states, as observed in captive iguanas and sea turtles [12,37].

In this rescued individual, the entry route for *B. bassiana* remains speculative but likely involved the aspiration of contaminated seawater. This interpretation aligns with the findings of Stockman et al. (2013), who reported that pneumonia in cold-stunned Kemp’s ridley turtles may develop as a consequence of sea water aspiration [48]. Their study further noted a higher prevalence of unilateral pulmonary abnormalities in the right lung compared to the left [48]. When both lungs were affected, the right lung tended to show more severe lesions, consistent with the presentation in this rescued individual. Anatomically, the entrance to the right main bronchus in *L. kempii* is located ventral to the entrance to the left main bronchus at the tracheal bifurcation, a feature that likely predisposes the right lung to more easily accumulate sea water during aspiration, and therefore to more severe pneumonia [48].

Histopathology revealed bronchial and interstitial pneumonia with septate hyphae. The presence of septate fungal elements, demonstrated by PAS staining, confirmed the infection by *B. bassiana*, which was subsequently cultured and identified via 18S and ITS rRNA sequencing. Notably, the fungus failed to grow at 37 °C [32], consistent with its optimal growth range of 25–30 °C, favouring colonisation in ectothermic hosts under cool conditions. Moreover, histopathology revealed a marked reduction in melanomacrophages in both the liver and spleen, indicating a potential impairment in the non-specific immune response. This, in combination with hepatocyte vacuolisation—likely due to lipid reserve depletion—supports the hypothesis of prolonged anorexia and emaciation, and higher predisposition to fungal colonisation and systemic dissemination.

*B. bassiana* was not isolated from the carapace lesions antemortem; instead, *Mucor* spp. were cultured from these external lesions. Mucormycete colonies are fast-growing, often covering the entire culture plate with dense, woolly mycelium. This prolific growth may have inhibited the development of other fungi in vitro, and thus the involvement of *B. bassiana* in the shell lesions cannot be ruled out [49]. Additionally, while *Mucor* spp. might have contributed to the pathogenesis of the carapace lesions, they are unlikely to be involved as agents of pneumonia, as the hyphae observed histologically were septate, rather than broad and coenocytic, characteristic of the order Mucorales.

Importantly, antifungal susceptibility testing, such as the E-test method, remains underutilised in sea turtle medicine, despite its critical role in guiding targeted therapy for fungal infections. The E-tests enable determination of minimum inhibitory concentrations (MICs) for antifungal agents, including voriconazole, itraconazole, amphotericin B, and posaconazole, drugs with potential clinical relevance in reptile and wildlife medicine. In the present study, the isolate demonstrated high resistance to fluconazole and amphotericin B (MICs > 256 µg/mL) but was susceptible to itraconazole (0.047 µg/mL) and voriconazole (0.004 µg/mL). These findings are consistent with previous reports, in which *B. bassiana* isolates demonstrated elevated MICs for fluconazole and amphotericin B, and lower MICs for itraconazole and voriconazole [12,15]. Given this pattern of resistance, empirical antifungal treatment without prior susceptibility data may result in therapeutic failure, particularly for pulmonary mycoses refractory to conventional antifungals. As such, incorporating E-tests or other standardised antifungal susceptibility methods into sea turtle rehabilitation protocols could significantly improve case outcomes, especially in fungal pneumonias refractory to standard treatments [12].

The concurrent bacterial isolates from pulmonary tissue—*Vagococcus fluvialis*, *Pseudomonas putida*, and *Citrobacter gillenii*—may have contributed to a polymicrobial infection [50]. Among these, only *V. fluvialis* was resistant to marbofloxacin, the antibiotic prescribed on the day of admission. While *P. putida* and *Citrobacter* spp. are recognised opportunists in sea turtles [51], *V. fluvialis* has not been previously reported as a pathogen in this context [52,53]. *V. fluvialis* is increasingly recognised as an emerging concern in clinical and One Health contexts, characterised by rising identification in human disease, occurrence across environmental and animal reservoirs, and accumulating antimicrobial resistance, so its co-isolation in this debilitated sea turtle is best interpreted as contextual information for integrated surveillance rather than evidence of species-specific emergence [54,55,56,57,58,59]. A limitation of this study is the absence of a metataxonomic screening of both bacterial (16s SSU rRNA gene amplicon sequencing) and fungal (e.g., ITS2 gene amplicon sequencing) communities, which would have showed a more comprehensive picture of the dysbiosis that favoured the fungal infection, potentially highlighting a greater diversity of secondary invaders.

Recent literature further supports a rising prevalence of fungal infections in sea turtles, particularly among Kemp’s ridley sea turtles. Mastrostefano et al. (2024) [12] reported that this species represented the majority of mycotic cases in their caseload, raising the possibility of a species-specific vulnerability. While this may be influenced by sampling bias, their findings emphasise the need for routine fungal screening in debilitated turtles, especially those presenting with skin or shell lesions, emaciation, or respiratory compromise.

This case underscores the complex interplay between climate, oceanography, and emerging pathogens, reinforcing the need for a One Health approach in sea turtle conservation. It also highlights the importance of including fungal aetiology—often underdiagnosed or mistaken for bacterial or viral causes—in the diagnostic workup of debilitated sea turtles, particularly when clinical deterioration persists despite antibiotic therapy. Integration of molecular diagnostics, culture-based methods, and antifungal susceptibility testing can guide targeted therapy, potentially improving rehabilitation outcomes and advancing wildlife health management.

## 6. Conclusions

Although rare, the occurrence of Kemp’s ridley sea turtles (*L. kempii)* in Portuguese waters provides valuable insights into the dispersal behaviour and oceanographic dynamics influencing this critically endangered species. Given the conservation status of Kemp’s ridley turtles, isolated rehabilitation cases can contribute meaningfully to conservation efforts, even when clinical outcomes are unfavourable. Current knowledge of the mycobiota of this species, as well as data on antifungal susceptibility of isolates obtained from affected individuals, remains scarce. The identification of *Beauveria bassiana* as the aetiological agent in this case emphasises the growing relevance of emerging fungal pathogens in sea turtle health and rehabilitation. Infections by *B. bassiana* typically occur under conditions of immunosuppression, suboptimal husbandry, or environmental stress, where the ectothermic nature of reptiles allows for fungal proliferation at lower body temperatures. From a One Health and conservation perspective, increasing awareness of mycotic diseases in marine reptiles is crucial, especially as climate change and environmental disruption continue to shift pathogen dynamics and host susceptibility in vulnerable wildlife populations.

## Figures and Tables

**Figure 1 microorganisms-13-02092-f001:**
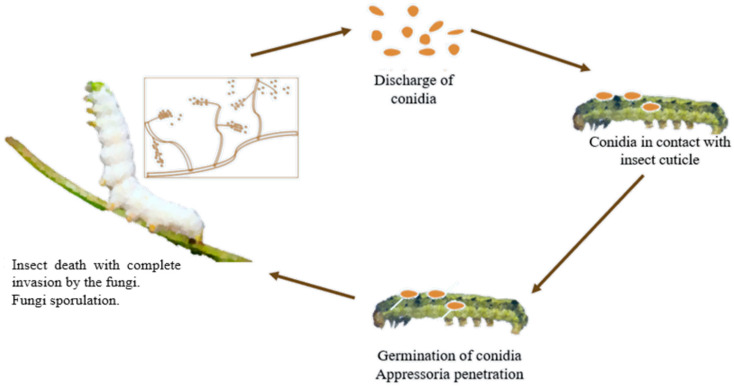
Life cycle of entomopathogenic fungi on insects.

**Figure 2 microorganisms-13-02092-f002:**
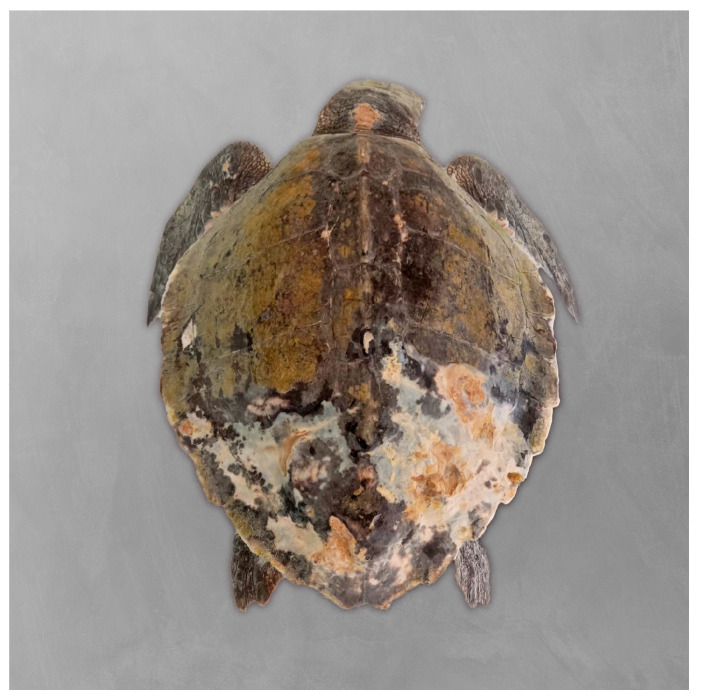
Kemp’s ridley sea turtle (*Lepidochelys kempii)* rescued off the coast of Sines, Portugal, and admitted to Porto d’Abrigo, Zoomarine, Portugal. Multiple lesions on the carapace.

**Figure 3 microorganisms-13-02092-f003:**
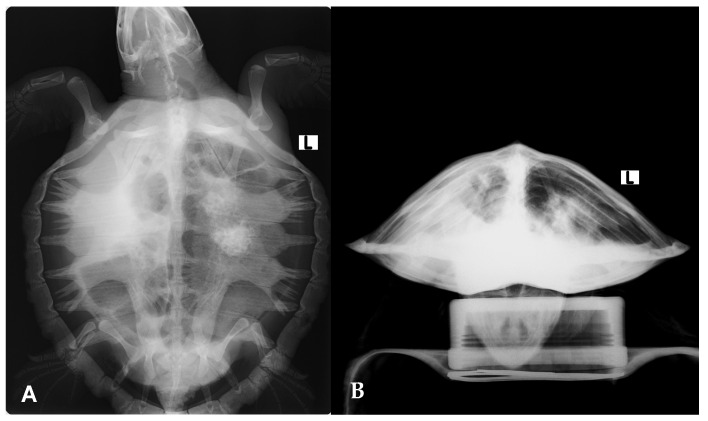
Radiographic evaluation of the Kemp’s ridley sea turtle (*Lepidochelys kempii)* shows increased pulmonary opacity, most pronounced in the right lung, along with ill-defined focal opacities in the left lung field. (**A**) Dorsoventral view; (**B**) craniocaudal view.

**Figure 4 microorganisms-13-02092-f004:**
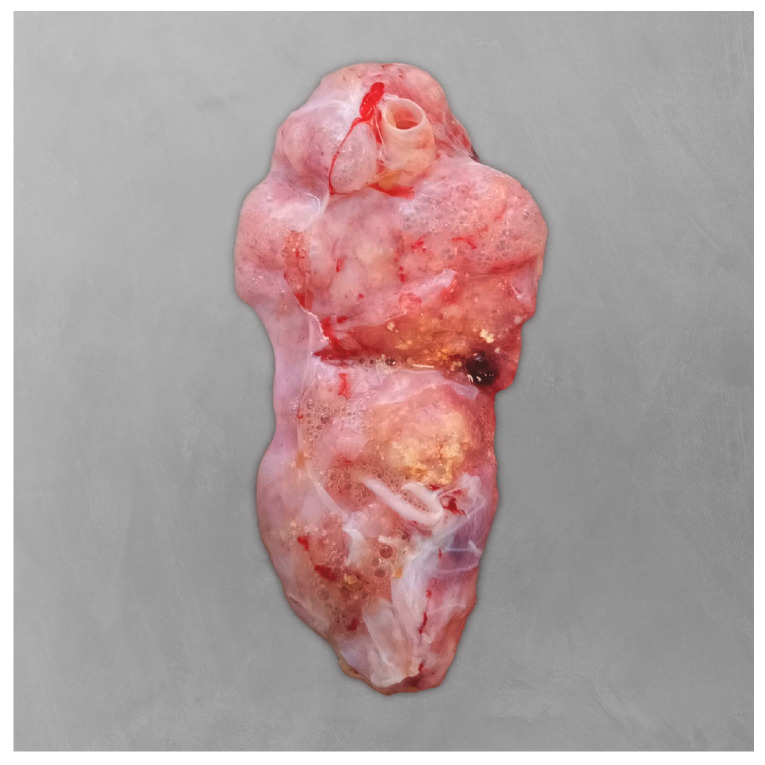
Postmortem examination of the right lung of the Kemp’s ridley sea turtle (*Lepidochelys kempii*). The lungs appeared diffusely oedematous, with multiple coalescing whitish granules.

**Figure 5 microorganisms-13-02092-f005:**
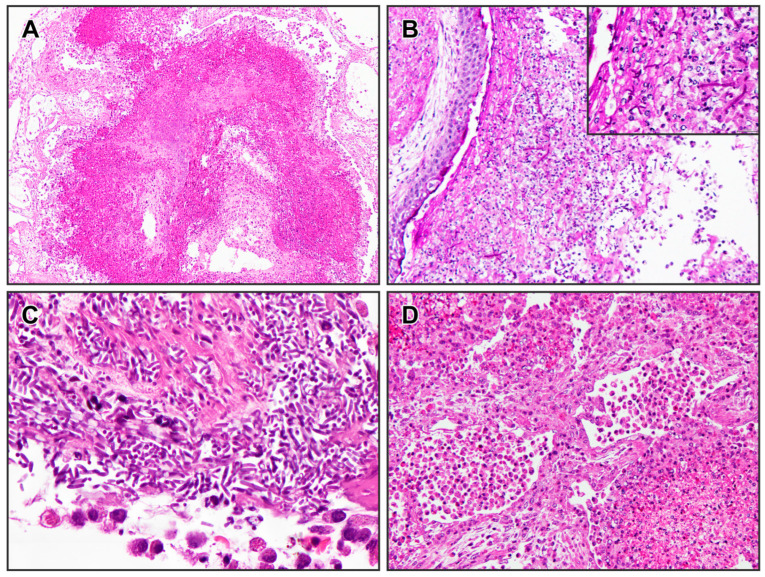
Lung histopathological findings from the Kemp’s ridley sea turtle (*Lepidochelys kempii*). (**A**) Severe bronchial and interstitial pneumonia (H&E, ×40). (**B**) In some bronchi, the exudate is less cellular and rich in fungal hyphae (PAS, ×100). Insert: hyphae structure within the exudate (PAS, ×400). (**C**) In some areas, the fungal hyphae are very numerous (H&E, ×400). (**D**) In areas around the bronchi, the faveoli have thickened walls with severe pneumocytes sloughing, eventually side by side with faveoli containing exudate, such as the one in the lower right side (H&E, ×100).

**Figure 6 microorganisms-13-02092-f006:**
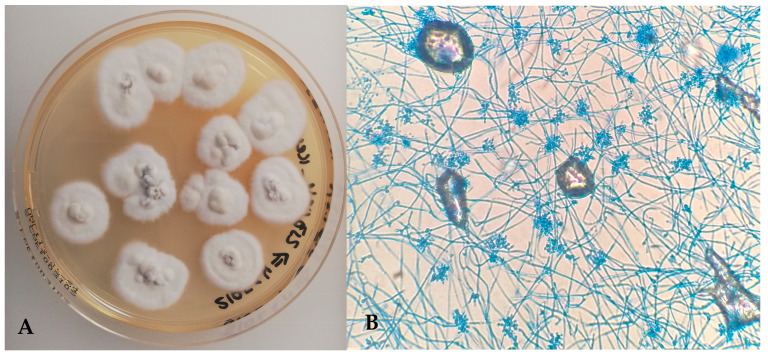
(**A**) *Beauveria bassiana*—cottony white fungal colonies on Sabouraud Dextrose Chloramphenicol Agar (25 °C); (**B**) narrow, septate hyaline hyphae and small conidia arranged in clusters (Lactophenol Cotton Blue, ×400).

**Table 1 microorganisms-13-02092-t001:** Reported cases of *Beauveria bassiana* infection in reptiles, detailing affected species, lesion types, and clinical outcomes.

Species	Observed Lesion	Outcome	Reference
American alligator (*Alligator mississippiensis*)	The thoracic cavity contained large masses of white, fluffy material. More than half of the pulmonary parenchyma was replaced by dark, consolidated tissue.	Deceased	[34]
Common house gecko (*Hemidactylus frenatus*)	Lesion on the forehead	Deceased	[35]
Galápagos giant tortoise (*Testudo elephantopus*)	The lung contained many small areas of tough, woody material.	Deceased	[36]
Green iguana (*Iguana iguana*)	Disseminated fungal infection	Deceased	[37]
Green sea turtle (*Chelonia mydas*)	Severe multifocal to coalescent granulocytic pulmonary abscesses with fibrinogranulocytic exudative pneumonia, focal ulcerative dermatitis of the flipper with bacterial and fungal hyphae	Deceased	[15]
Kemp’s ridley sea turtle (*Lepidochelys kempii*)	Severe granulocytic and granulomatous pneumonia with intralesional fungal hyphae, vascular invasion, and necrosis; coelomitis with granulocytic granuloma formation and intralesional fungal hyphae in multiple organs; severe multifocally extensive granulocytic nephritis with intralesional fungal hyphae	Deceased	[15]
	Pneumonia, with pleuritis	Deceased	[12]
	Pneumonia	Deceased	[12]
	Pneumonia, hepatitis	Deceased	[12]
	Pneumonia, with pleuritis; necroulcerative enteritis	Deceased	[38]
Leatherback sea turtle (*Dermochelys coriacea*)	Pneumonia	Deceased	[12]
Loggerhead sea turtle (*Caretta caretta*)	Chronic pyogranulomatous pneumonia with a lymphocytic–plasmacytic component	Deceased	[15]
Red-eared slider (*Trachemys scripta*)	Pulmonary congestion with pleuritis and presence of yellowish nodules	Deceased	[39]
Nikolsky’s viper (*Vipera berus nikolskii*)	Skin lesion	Survived	[40]

## Data Availability

The original contributions presented in this study are included in the article. Further inquiries can be directed to the corresponding author.

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
