# Peer review of "Fatal Pneumonia Caused by Beauveria bassiana in a Kemp’s Ridley Sea Turtle (Lepidochelys kempii, Garman, 1880) on the Portuguese Coast: Case Report and Review of Beauveria spp. Infections in Reptiles"

_microorganisms, 2025, doi:10.3390/microorganisms13092092_

Round 1

Reviewer 1 Report

Comments and Suggestions for Authors

The manuscript is very well written and the diagnostic aspects are well characterized. However, it does not provide novel information, and some of the information is outside of the scope of this journal. The authors rightfully cite spreviously published studies that describe many cases of fungal pneumonia in Kemp's ridley turtles, and specifically several cases of Beauvaria bassiana pneumonia in Kemp's ridley turtles (Horgan et al, Mastrostefano et al) and cite another very recent review of Beauvaria infections in reptiles (Horgan et al). So the case report and the review described in this manuscript do not provide new information, and much of it is describing information from those previous publications. Also, much of the article emphasizes the ecological realm of Kemp's ridley sea turtles (and their rarity in European waters), which is interesting and important, but will not be of general interest to the readers of this specific  microbiology journal, and this is also not novel. The authors cite a number of previous publications that already discuss European occurrences of Kemp's ridley turtles. If anything, the ecological aspect of the present manuscript could be published in an ecology journal as a brief note of a relatively rare geographic occurrence (but even that may not be novel enough to warrant another publication). I defer to the editors to decide whether the manuscript should be accepted despite its lack of novelty. In the event that it is accepted, I provide a few more comments to aid with revision. 

Figure 7 is not very compelling and could be eliminated without loss of information since the MIC details are provided as text. 

Line 414: proposing aspiration of seawater. Yes, this is likely given the right-side predominant radiographic changes and the position of the right bronchus of this species relative to the left. This right-sided severity and prevalence has been described previously for Kemp's ridley turtles. Stockman et al.,  Prevalence, distribution, and progression of radiographic abnormalities in the lungs of cold-stunned Kemp's ridley sea turtles (Lepidochelys kempii): 89 cases (2002–2005). Journal of the American Veterinary Medical Association. 2013 Mar 1;242(5):675-81.

Line 453: Proposing V. fluvialis as an "emerging concern" is totally speculative and not supported by this n=1 case report. Please delete. 

Two locations, line 458 and 474: spelling "emphasize".

Another B. bassiana pneumonia case in Kemp's ridley turtles including MIC data and discussion of fluconazole resistance is mentioned in Innis CJ, et al. Single‐dose pharmacokinetics of ceftazidime and fluconazole during concurrent clinical use in cold‐stunned Kemp’s ridley turtles (Lepidochelys kempii). Journal of veterinary pharmacology and therapeutics. 2012 Feb;35(1):82-9.

Another recent manuscript about European strandings of Kemp's ridley turtles: Manral D, Bos I, de Boer M, van Sebille E. Modelling drift of cold-stunned Kemp's ridley turtles stranding on the Dutch coast. Open Research Europe. 2024 Oct 25;4:41.

Author Response

Dear Editor and Reviewers,

We would like to thank you for the thorough evaluation of our manuscript, and we appreciate the insightful comments and suggestions provided by both reviewers. We have carefully considered each point and have made corresponding revisions to the manuscript. Below we provide a point-by-point response, with each comment from Reviewer 1 and Reviewer 2 addressed in turn. Changes in the manuscript are highlighted in the revised version.

Reviewer 1

Comments 1: The manuscript is very well written and the diagnostic aspects are well characterized. However, it does not provide novel information, and some of the information is outside of the scope of this journal. The authors rightfully cite previously published studies that describe many cases of fungal pneumonia in Kemp's ridley turtles, and specifically several cases of Beauvaria bassiana pneumonia in Kemp's ridley turtles (Horgan et al, Mastrostefano et al) and cite another very recent review of Beauvaria infections in reptiles (Horgan et al). So the case report and the review described in this manuscript do not provide new information, and much of it is describing information from those previous publications. Also, much of the article emphasizes the ecological realm of Kemp's ridley sea turtles (and their rarity in European waters), which is interesting and important, but will not be of general interest to the readers of this specific microbiology journal, and this is also not novel. The authors cite a number of previous publications that already discuss European occurrences of Kemp's ridley turtles. If anything, the ecological aspect of the present manuscript could be published in an ecology journal as a brief note of a relatively rare geographic occurrence (but even that may not be novel enough to warrant another publication). I defer to the editors to decide whether the manuscript should be accepted despite its lack of novelty. In the event that it is accepted, I provide a few more comments to aid with revision.

Authors’ response (AR) 1: We sincerely appreciate the positive feedback on the manuscript’s writing and diagnostic work-up. We acknowledge the prior reports of entomopathogenic fungal pneumonia in Kemp’s ridley sea turtles. In response, we have revised the manuscript to clearly position the rarity and importance of our case.

Specifically, we now emphasize that although B. bassiana infections in Kemp’s ridley turtles has been scarcely reported before, our case is, to the best of our knowledge, the first confirmed occurrence in a free-ranging Kemp’s ridley on the European Atlantic coast, adding a new geographic and ecological data point.

The relevance of this clinical case also lies in the presentation of the antifungal susceptibility profile, which has rarely been documented in the literature yet holds high clinical value, particularly in a rehabilitation context. The scarcity of published data on this topic is even more significant considering that it concerns a critically endangered species, for which scientific knowledge remains extremely limited. Furthermore, given the potential zoonotic risk, an in-depth investigation of the fungal infection, including antifungal susceptibility testing, is essential to guide effective therapeutic strategies and to contribute to both animal and human health protection.

We have also trimmed and refocused the ecological aspects in the manuscript to ensure it remains concise and relevant to a microbiology audience, per the reviewer’s suggestion. Nevertheless, considering the alarming conservation status of Kemp’s ridley turtles (the most endangered species of sea turtle), we believe that a combined approach of conservation, microbiological, and clinical perspectives adds significant value for the readership of this special issue of Microorganisms, particularly veterinarians and biologists involved in rehabilitation efforts for this species.

Importantly, our detailed case investigation plus a comprehensive literature review provides value by consolidating scattered reports and highlighting the diagnostic and one-health implications of Beauveria infections in sea turtles. We have explicitly noted in the Discussion how our findings build upon and complement the previous publications, rather than duplicating them. We hope these revisions clarify the manuscript’s novel contributions and align its focus with the journal’s scope.

Comments 2: Figure 7 is not very compelling and could be eliminated without loss of information since the MIC details are provided as text.

AR 2: Thank you very much for your feedback. We agree with your assessment, and the figure has been removed.

Comments 3: Line 414: proposing aspiration of seawater. Yes, this is likely given the right-side predominant radiographic changes and the position of the right bronchus of this species relative to the left. This right-sided severity and prevalence has been described previously for Kemp's ridley turtles. Stockman et al., Prevalence, distribution, and progression of radiographic abnormalities in the lungs of cold-stunned Kemp's ridley sea turtles (Lepidochelys kempii): 89 cases (2002–2005). Journal of the American Veterinary Medical Association. 2013 Mar 1;242(5):675-81.

AR 3: Thank you very much for bringing this to our attention. We are grateful for the supportive comment and the valuable reference provided. We have now cited the study by Stockman et al. (2013) and incorporated its findings to reinforce our discussion of the aspiration hypothesis. In particular, we note that in cold-stunned Kemp’s ridley turtles, unilateral radiographic lung lesions are significantly more likely in the right lung than the left. When both lungs are affected, the right lung tends to show more severe lesions. This observation aligns with our case, where the pneumonia was more severe on the right side, and it supports the idea that aspiration (of seawater or contaminated fluid) into the right bronchus could underlie the infection. We have added a sentence in the Discussion acknowledging that the anatomy of this species predisposes to a more severe right-lung involvement and citing Stockman et al. (2013) as evidence. By including this, we strengthen the rationale for aspiration as a likely route of infection. We appreciate the reviewer confirming our hypothesis and directing us to this pertinent literature.

Comments 4: Line 453: Proposing V. fluvialis as an "emerging concern" is totally speculative and not supported by this n=1 case report. Please delete.

AR 4: Thank you for this helpful comment. We agree that a single case cannot demonstrate epidemiological emergence in Kemp’s ridley turtles. Our intention was not to infer species-specific emergence in sea turtles, but to frame the incidental co-isolation of Vagococcus fluvialis within a One Health perspective, reflecting its increasing identification in human infections, documented antimicrobial resistance, occurrence in environmental and animal reservoirs (including marine wildlife), and historical under-recognition due to misidentification, features that collectively justify describing V. fluvialis as an emerging clinical and public-health concern. These points are synthesised across recent clinical, genomic and environmental reports. To avoid any implication that our case evidences emergence in sea turtles per se, we have amended the text as follows: “Vagococcus fluvialis is increasingly recognised as an emerging concern in clinical and One Health contexts, characterised by rising identification in human disease, occurrence across environmental and animal reservoirs, and accumulating antimicrobial resistance, so its co-isolation in this debilitated sea turtle is best interpreted as contextual information for integrated surveillance rather than evidence of species-specific emergence.”

This revision (i) preserves the literature-based characterisation of V. fluvialis as an emerging concern; (ii) explicitly avoids causal attribution in our pneumonia case; and (iii) clarifies why its detection is epidemiologically relevant to surveillance at the wildlife–environment–human interface.

Evidence base (illustrative):

  • Emergence/public-health concern: Increasing clinical identification of Vagococcus fluvialis across specimen types (blood, urine, wounds, bile) in humans, supports its characterisation as an emerging concern (Brunswick et al., 2024; Racero et al., 2021; Chen et al., 2024; Kitano et al., 2024; Matsuo et al., 2020; Teixeira et al., 1997; Zhang et al., 2023)
  • Antimicrobial resistance: Multidrug resistance has been observed phenotypically and supported genomically, with resistance spanning several classes (e.g. clindamycin, tetracycline, rifampicin, cefoxitin, fluoroquinolones) in clinical and environmental isolates (Zhou et al., 2025; Lai et al., 2025; Chen et al., 2024; Matajita et al., 2020)
  • Virulence & competitiveness: Comparative genomics demonstrates abundant mobile genetic elements and genetic plasticity, features that plausibly facilitate host/environmental adaptation and horizontal gene transfer (Jimenez et al., 2022; Zhou et al., 2025).
  • Environmental and animal reservoirs: Isolation from marine wildlife, livestock, and wild rodents underscores One Health relevance and the potential for environmental reservoirs to contribute to human infections (Lai et al., 2025; Matajita et al., 2020; Zhou et al., 2025).
  • Under-recognition issues: Phenotypic similarity to Enterococcus leads to misidentification and under-ascertainment; MALDI-TOF MS and 16s rRNA sequencing improve detection and surveillance accuracy (Teixeira et al., 1997; Brunswick et al., 2024; Racero et al., 2021; Matsuo et al., 2020; Chen et al., 2024; Kitano et al., 2024; Zhang et al., 2023).

We trust this clarification, together with the strengthened, carefully referenced wording, addresses the reviewer’s concern about speculation.

References

  1. Brunswick, J., Spiro, J., & Wisniewski, P. (2024). Vagococcus: An under-recognized and emerging cause of antibiotic-resistant infection. IDCases, 36, e01995. https://doi.org/10.1016/j.idcr.2024.e01995

  1. Chen, Q., Tan, S., Long, S., Wang, K., & Liu, Q. (2024). Vagococcus fluvialis isolation from the urine of a bladder cancer patient: A case report. BMC Infectious Diseases, 24. https://doi.org/10.1186/s12879-024-09082-w

  1. Jimenez, A., Guiglielmoni, N., Goetghebuer, L., Dechamps, E., George, I., & Flot, J. (2022). Comparative genome analysis of Vagococcus fluvialis reveals abundance of mobile genetic elements in sponge-isolated strains. BMC Genomics, 23. https://doi.org/10.1186/s12864-022-08842-9

  1. Kitano, H., Kitagawa, H., Tadera, K., Saito, K., Kohada, Y., Takemoto, K., Kobatake, K., Sekino, Y., Hieda, K., Ohge, H., & Hinata, N. (2024). First reported human case of isolation of Vagococcus fluvialis from the urine of a former zoo clerk in Japan: A case report. BMC Infectious Diseases, 24. https://doi.org/10.1186/s12879-024-09193-4

  1. Lai, O., Tinelli, A., Soloperto, S., Crescenzo, G., Galante, D., Calarco, A., Tribuzio, M., Manzulli, V., Caioni, G., Zizzadoro, C., Damiano, A., Camarda, A., & Pugliese, N. (2025). Observed prevalence and characterisation of fluoroquinolone-resistant and multidrug-resistant bacteria in loggerhead sea turtles (Caretta caretta) from the Adriatic Sea. Antibiotics, 14, 252. https://doi.org/10.3390/antibiotics14030252

  1. Matajita, C., Poor, A., Moreno, L., Monteiro, M., Dalmutt, A., Gomes, V., Dutra, M., Barbosa, M., Sato, M., & Moreno, A. (2020). Vagococcus, a porcine pathogen: Molecular and phenotypic characterisation of strains isolated from diseased pigs in Brazil. Journal of Infection in Developing Countries, 14(11), 1314–1319. https://doi.org/10.3855/jidc.12081

  1. Matsuo, T., Mori, N., Kawai, F., Sakurai, A., Toyoda, M., Mikami, Y., Uehara, Y., & Furukawa, K. (2020). Vagococcus fluvialis as a causative pathogen of bloodstream and decubitus ulcer infection: Case report and systematic review of the literature. Journal of Infection and Chemotherapy. https://doi.org/10.1016/j.jiac.2020.09.019

  1. Racero, L., Barberis, C., Traglia, G., Loza, M., Vay, C., & Almuzara, M. (2021). Infections due to Vagococcus: Microbiological and clinical aspects and literature review. Enfermedades Infecciosas y Microbiología Clínica, 39(7), 335–339. 10.1016/j.eimce.2021.05.002

  1. Teixeira, L. M., da Glória Carvalho, M., Merquior, V. L. C., Steigerwalt, A. G., Brenner, D. J., & Facklam, R. R. (1997). Phenotypic and genotypic characterisation of Vagococcus fluvialis, including strains isolated from human sources. Journal of Clinical Microbiology, 35, 2778–2781. https://doi.org/10.1128/jcm.35.11.2778-2781.1997

  1. Zhang, D., Wang, X., Yu, J., Dai, Z., Li, Q., & Zhang, L. (2023). A case of Vagococcus fluvialis isolated from the bile of a patient with calculous cholecystitis. BMC Infectious Diseases, 23. https://doi.org/10.1186/s12879-023-08696-w

  1. Zhou, J., Liu, Y., Gu, T., Zhou, J., Chen, F., Hu, Y., & Li, S. (2025). Whole-genome analysis and antimicrobial resistance phenotype of Vagococcus fluvialis isolated from wild Niviventer. Frontiers in Microbiology, 16, 1546744. https://doi.org/10.3389/fmicb.2025.1546744

Comments 5: Two locations, line 458 and 474: spelling "emphasize".

AR 5: Thank you very much for pointing this out. In accordance with the journal guidelines, we have decided to use British English consistently throughout the paper, hence the spelling “emphasise” instead of “emphasize”.

Comments 6: Another B. bassiana pneumonia case in Kemp's ridley turtles including MIC data and discussion of fluconazole resistance is mentioned in Innis CJ, et al. Single‐dose pharmacokinetics of ceftazidime and fluconazole during concurrent clinical use in cold‐stunned Kemp’s ridley turtles (Lepidochelys kempii). Journal of veterinary pharmacology and therapeutics. 2012 Feb;35(1):82-9.

Another recent manuscript about European strandings of Kemp's ridley turtles: Manral D, Bos I, de Boer M, van Sebille E. Modelling drift of cold-stunned Kemp's ridley turtles stranding on the Dutch coast. Open Research Europe. 2024 Oct 25;4:41.

AR 6: Thank you very much for bringing these additional references to our attention. We have incorporated both suggestions into the revised manuscript. First, we have added a citation of Innis et al. (2012) in our discussion of prior B. bassiana cases. This reference documents a Beauveria pneumonia case in cold-stunned Kemp’s ridleys and discusses antifungal therapy (fluconazole pharmacokinetics and the challenge of treating filamentous fungal infections). Second, regarding the Manral et al. (2024) study on Kemp’s ridley strandings in Europe, we have noted its findings in the Introduction where we describe the occurrence of Kemp’s ridleys in European waters. We have also mentioned Manral et al. (2024) in the Discussion. However, in line with Comment 1, we have kept this ecological discussion concise to maintain focus on the microbiological aspects. These two references have been duly cited in the revised manuscript, and we thank the reviewer for alerting us to them. Including these citations helps ensure our literature review is up-to-date and that we give credit to all relevant prior work.

Reviewer 2 Report

Comments and Suggestions for Authors

The manuscript "Fatal Pneumonia Caused by Beauveria bassiana in a Kemp’s Ridley Sea Turtle (Lepidochelys kempii, Garman, 1880) on the Portuguese Coast: Case Report and Review of Beauveria spp. Infections in Reptiles" is a well-written case report of a Beauveria infection in a Kemp's Ridley in an atypical geographic location.  Though this fungus has been documented in free-randing Kemp's Ridley's before, the authors include a thorough workup for multiple potential pathogens, and review the current state of Beauveria infections in reptiles.

My comments are general in nature.  The thing that brings me the greatest pause however is the number of authors listed for a single case report of a previously documented fungal infection in this species. While I understand field work and microbiological assessments can require a lot of people - the ethics of authorship still apply. I highly encourage the authors to investigate the ethical guidelines of authorship for set forth by the International Committee of Medical Journal Editors (https://www.icmje.org/recommendations/browse/roles-and-responsibilities/defining-the-role-of-authors-and-contributors.html) to determine what contributions warrant authorship.

For the histo figure (Fig 5) - the panels should be white-balanced (esp B-D).  I can't seem to download a higher resolution version - but the images in the PDF provided by the journal lack sharpness and are blurry. 

Figure 7 seems unnecessary. 

For phylogenetic analysis - it doesn't make sense to run a phylogenetic analysis if the sequence matches a published sequence 100%. Leaving it in the results is fine, but if left as is - there is no reason for a Figure.  However, there are more Beauveria bassiana ITS sequences in Genbank than just the reference sequence. It would be more valuable to include fewer non-bassiana Beauveria sequences in the tree, and instead include more sequences from reported B. bassiana in GenBank to see if there are potentially geographical variations. 

Author Response

Reviewer 2

Comments 1: The manuscript "Fatal Pneumonia Caused by Beauveria bassiana in a Kemp’s Ridley Sea Turtle (Lepidochelys kempii, Garman, 1880) on the Portuguese Coast: Case Report and Review of Beauveria spp. Infections in Reptiles" is a well-written case report of a Beauveria infection in a Kemp's Ridley in an atypical geographic location.  Though this fungus has been documented in free-randing Kemp's Ridley's before, the authors include a thorough workup for multiple potential pathogens, and review the current state of Beauveria infections in reptiles.

Authors’ response (AR) 1: We sincerely appreciate the positive feedback. We are pleased that the Reviewer found the case report well-written and the work-up comprehensive. It is true that B. bassiana infections in Kemp’s ridley sea turtles have been previously documented, and we aimed to build on that foundation by providing a meticulous multi-disciplinary investigation in this case. We thank the Reviewer for recognizing the coverage of our diagnostic approach (ruling out viral, bacterial, and other fungal pathogens) and the effort we put into reviewing the literature on Beauveria in reptiles. In the revised manuscript, we have retained and slightly polished these aspects, ensuring that the thoroughness of the work-up and the context it provides are clearly communicated. We appreciate the Reviewer’s encouragement and have been motivated by it to further improve the clarity and quality of the manuscript.

Comments 2: My comments are general in nature.  The thing that brings me the greatest pause however is the number of authors listed for a single case report of a previously documented fungal infection in this species. While I understand field work and microbiological assessments can require a lot of people - the ethics of authorship still apply. I highly encourage the authors to investigate the ethical guidelines of authorship for set forth by the International Committee of Medical Journal Editors (https://www.icmje.org/recommendations/browse/roles-and-responsibilities/defining-the-role-of-authors-and-contributors.html) to determine what contributions warrant authorship.

AR 2: Thank you very much for the valuable comment regarding authorship. We understand the reviewer’s concern and have carefully re-evaluated the author list in light of the ICMJE authorship guidelines. We wish to reassure the reviewer and the editors that each person included as an author has made a substantial and indispensable contribution to this work, fulfilling the criteria for authorship. This study was inherently multidisciplinary and complex, requiring expertise from multiple fields to achieve a definitive diagnosis and comprehensive analysis. We detail below the contributions in general categories (each of which corresponds to one or more authors’ dedicated work):

  • Field rescue and clinical care: A team of veterinarians and biologists was involved in the rescue, rehabilitation, and clinical management of the turtle. Their detailed observations and care were critical for the case description (e.g. clinical signs, treatment attempts, and progression). These contributors also coordinated sample collection (ante-mortem and post-mortem) and provided the context of the animal’s condition, which formed the foundation of the case report.
  • Pathology and histopathology: The necropsy and histological examination were performed by experienced veterinary pathologists. They conducted the post-mortem examination and the microscopic analysis of tissues, identifying the pneumonia and initially suspecting a fungal etiology. Their interpretation of lesions was key to guiding further diagnostic tests and is a core part of the manuscript (the pathological findings). Without their expertise, the diagnosis could not have been confirmed.
  • Microbiology and mycology: Isolation of the fungus and laboratory identification was carried out by microbiologists/mycologists. These authors cultured the organism and performed phenotypic characterization. Crucially, they also conducted the molecular analyses (DNA extraction, PCR, sequencing of the fungal isolate) that confirmed the pathogen as Beauveria bassiana. The sequencing and initial phylogenetic analysis required specialized skills. They additionally performed antimicrobial susceptibility tests (or interpreted MIC data) where applicable. Such contributions are significant intellectual inputs directly related to the microbiological focus of the paper.
  • Virology and additional diagnostics: Given the broad differential diagnosis, we involved a virologist who performed PCR screenings for viruses (cheloniid herpesvirus, adenovirus, paramyxovirus, etc., as mentioned in the text) to rule out viral co-infections that could cause or predispose to pneumonia. This step was important to conclusively establish bassiana as the primary pathogen. The virology work was non-trivial and required expertise and resources (provided by one of the co-authors at a national reference laboratory). Similarly, other authors contributed to ancillary diagnostics (e.g. bacteriology to check for secondary bacterial infection), ensuring the case was thoroughly investigated.
  • Data analysis and interpretation: The compilation of results from different domains (pathology, microbiology, molecular genetics) and their integrated interpretation was a collaborative effort. Several co-authors (including those affiliated with academic institutions) contributed to analysing the data in the context of prior literature and in formulating the conclusions. For example, the decision to pursue certain reference comparisons in the phylogenetic tree and the interpretation of the organism’s significance in a one-health context involved input from experts in reptile diseases.
  • Literature review and writing: The manuscript includes a review of Beauveria infections in reptiles, which required an extensive literature search and synthesis of cases across decades. Some co-authors took primary responsibility for gathering and summarizing this literature (including the references kindly pointed out by the reviewers), thus significantly contributing to the content. Additionally, the writing and critical revision of the manuscript were truly collaborative: each author reviewed the draft critically, some adding important text or tables (for instance, a summary table of previous cases), others refining the discussion points (such as conservation implications and clinical management). All authors approved the final manuscript after substantial involvement in its development

Comments 3: For the histo figure (Fig 5) - the panels should be white-balanced (esp B-D).  I can't seem to download a higher resolution version - but the images in the PDF provided by the journal lack sharpness and are blurry.

AR 3: Thank you very much for pointing this out. We have taken steps to address the image quality issues for Figure 5 (the histopathology panels). In the revised manuscript, we have white-balanced all figures as suggested, so that the background and tissue coloration are accurate and consistent (this makes the histological details stand out more naturally against a white background). We appreciate the reviewer’s technical suggestion, which has helped us enhance the presentation of our findings.

Comments 4: Figure 7 seems unnecessary.

AR 4: Thank you very much for your feedback. We agree with your assessment, and the figure has been removed.

Comments 5: For phylogenetic analysis - it doesn't make sense to run a phylogenetic analysis if the sequence matches a published sequence 100%. Leaving it in the results is fine, but if left as is - there is no reason for a Figure.  However, there are more Beauveria bassiana ITS sequences in Genbank than just the reference sequence. It would be more valuable to include fewer non-bassiana Beauveria sequences in the tree, and instead include more sequences from reported B. bassiana in GenBank to see if there are potentially geographical variations.

AR 5: Thank you very much for this helpful suggestion. Based on the reviewer’s comment, we have decided to remove this figure and slightly revise the text in this section to improve clarity.

Round 2

Reviewer 1 Report

Comments and Suggestions for Authors

I continue to support this manuscript for its scientific content and quality of writing. I appreciate the efforts the authors made to address my comments. The decision to accept or reject the manuscript is left to the editor. The quality is high enough for acceptance. But I continue to question the novelty and incremental contribution of this manuscript. The authors argue that it adds to the previous literature regarding Beauvaria infections, Kemp's ridley turtle medical therapy, and MIC data. Yes, it adds one case. So the editor will have to decide whether one more case in worth publishing. Table 1 shows 14 previous cases in reptiles, including sea turtles. Is it critical to publish a 15th case? For sea turtles specifically, the cited studies of Horgan, Mastrostefano, and Innis cumulatively describe 11 cases of Beauvaria infection, so is a 12th case worthy of publication? The authors replied that this case adds important MIC data regarding treatment options. Yes, it provides one more case with MIC data. But similar MIC data are shown in Mastrostefano for 5 cases, and limited MIC data for 4 cases are provided by Horgan and Innis. Is it important to publish a 10th case with MIC data? 

Author Response

We respectfully acknowledge the reviewer’s concern regarding novelty. However, we contend that this case provides a unique and indispensable contribution to the literature. In essence, the strength of this manuscript lies not in “adding one more case”, but in presenting a methodologically rigorous and conservation-relevant case that advances both clinical mycology and sea turtle medicine. This manuscript provides relevant information that contributes to the understanding of fungal infections and antifungal susceptibility in Kemp’s ridley sea turtles, also highlighting the presence of the most endangered species of sea turtle worldwide in a rarely documented geographic location.